# Hybrid RF Propagation Model using ITM and Gaussian Processes for Communication-Aware Planning

Spencer Watza, Ramya Kanlapuli Rajasekaran, and Eric Frew
Ann and H.J. Smead Aerospace Engineering Sciences
University of Colorado Boulder; Boulder, Colorado 80309
Email: spencer.watza@colorado.edu; ramya.kanlapulirajasekaran@colorado.edu; eric.frew@colorado.edu

*Abstract*—**Communication-aware robotics is a necessary step to successfully implement multi-agent missions with UAS. While distributed systems have been investigated, previous research focused on using simple RF propagation models or ignored it with assumptions. To develop higher understandings of the affects of wireless communication between agents, higher fidelity models are required. This work proposes a hybrid approach for simulation and in-situ mapping of the Path Loss fields. The architecture combines a prediction step using traditional propagation models with a learning step to correct errors. A version of this architecture is used for simulation and shows the plausibility of using Gaussian Processes to find the structure of an RF with sparse measurements.**

## I. INTRODUCTION

Using multiple robotic agents increases the mission possibilities with small UAS (sUAS). Industry and academics are interested in using teams of sUAS to better solve missions like remote sensing, package delivery, and target tracking [8], [15], [7], [14]. A key issue is that the agents must maintain links between each other while solving the task in order to gain benefits over a single agent systems [3], [16]. This requires the agents to be Communication-aware with their cooperative partners (Figure 1).

A crucial part of communication awareness is being able to estimate the link between nodes in order to avoid drop out or to reconnect with the network if a failure occurs [6]. In cooperative missions sharing information is crucial and with stronger links the information relay is more reliable. Current research that has investigated distributed autonomous systems have either assumed the links are maintained or used simple transmission equations in simulation. These assumptions are not reasonable for all operational environments, especially in complex environments. For example, mountainous terrain provides trouble to emergency workers and military forces when communicating between teams due to the frequency of large elevation changes and obstructions [1].

While there are dozens of propagation models and types that scientists have been developing for almost an entire century, there is no clear best model [9]. Many models are developed for a set of frequencies in specific environments like indoor or urban propagation [10]. Surprisingly, there has not been significant research recently in rural propagation due to the belief that it has already been solved, although Phillips et al.

show otherwise. We aim to bridge the gap between robotics simple RF propagation modeling with that used in communication focused work. This work proposes a hybrid architecture with traditional propagation models and machine learning to act as both a simulation environment and in-situ estimation for Communication-aware applications. Before diving into the proposed architecture, a brief review of RF propagation basics will be performed. The proposed architecture model will be presented in section II. The simulation environment and evaluations will be setup in Section III with their results will be discussed in section IV. The paper will conclude with a summary and planned future work in sections V and VI respectively.

### RF Propagation Basics

In wireless communication information is modulated into an RF signal that is transmitted through space. A receiver picks up this signal and attempts to obtain the information encoded in the signal. There are a variety of modulation schemes and error coding correction techniques that are used for the digital-analog portions of communication and will not be a focus here. The challenge in wireless channels is the receiver is also picking up signals from other sources or delayed signals from the desired transmitter which creates interference. In addition, the energy of the desired signal dissipates as it propagates and can be reflected, refracted, and scattered by objects in space. The ability for a receiver to piece together the information is directly related to how strong the main signal is when compared to the noise it is hearing; signal-to-noise ratio.

Wireless communication often doesn't use a single frequency for its signal but a bandwidth of frequencies centered around the carrier frequency. This bandwidth helps contain the information that is being sent. When a wireless signal attenuates, there are two main types of fading that can occur; frequency selective fading and flat fading. Frequency selective fading occurs when only a portion of the frequencies in the bandwidth attenuate while the rest are unaffected. In flat fading, all frequencies are faded. Flat fading can be assumed as a worst scenario as there are possibilities of recovering information from frequency fading by use of error correction.

The total fade of a signal is often referred to the Path Loss of a signal which is caused by free space loss, shadowing, and

multipath. Free space loss and shadowing are both characteristically slow which means that the change in the amount of loss is not dependent on time. Free space loss is energy dissipation which is a function of the distance between the transmitter and receiver while shadow fading is from obstructions in the path of the signal often hills or buildings (hence the term shadow). Multipath fading is characteristically fast which looks like random noise and is highly time dependent but usually has a smaller overall effect. Multipath is when multiple signals arrive at the receiver at the same time from different paths causing wave interference.

In this work Path Loss is referred to being free space loss and shadowing as they are primarily functions of positions. Multipath will be treated as an additional noise term as it is correlated to motion, antenna, location, and temperature among others. For additional reading material for RF propagation readers can explore books on wireless channels such as Rappaport [11]

## II. MODEL

The proposed architecture to model the RF propagation losses in-situ is a hybrid of traditional and active methods. The architecture consists of two components; an initial prediction of the mean field and a correction mean field. The two of these combined will provide the estimated mean field for the Communication-aware application. The goal is to take the advantages of both types of RF propagation models. Traditional models do an okay job at predicting path losses for a given environment but will have errors, however they can be used before an operation to provide a base line prediction. Active methods require sampling live to "learn" the field through some sort of model parameterization. This process can take time to sample and process which may not be practical before the sUAS operation begins [10]. The implementation in this paper uses a terrain model and a Gaussian Process for the active model.

In addition, this architecture is extensible for simulation as the terrain model will provide location based structure to the Path Loss. By adding a correcting field derived from flight tests or random processes, a higher fidelity model can be generated. This section will discuss the in-situ use of the two components.

### Prediction Step

Terrain models make predictions on the shadowing and free space loss effects using Digital Elevation Maps (DEM) which are available from the USGS. This allows for larger extensibility than most other types of models. The two main terrain models are the Irregular Terrain Model (ITM) also known as the Longley-Rice Model and the International Telecommunications Union Terrain model (ITU Terrain and ITU-R). SPLAT!, an open source software, was chosen to perform the prediction step which implements a modified version of ITM. SPLAT! can provide point to point analysis or area coverage for Path Loss including line of sight and average terrain elevations for the area. In the United States, Space Radar Topography Mission (SRTM) DEM data is available at high 1 degree arc

second resolution and lower 3 degree arc second resolution. For more information about SPLAT! and ITM, the authors refer the readers to http://www.qsl.net/kd2bd/splat.html and Hufford [4].

Since SPLAT! is calculated at every pixel, it is reasonable to make the Path Loss field discrete. A grid structure is created centered about the region of interest and finds the value of the cell by averaging all SPLAT! points that fall in it. The limitation of this method is the resolution of the DEM, however may be fine for operation due to errors in GNC. There are different techniques to generate a more continuous field from the SPLAT! output but would make generating and planning on this information computationally intractable.

### Correction Step

Using the same grid structure that was generated for the prediction step, a correction is generated for each cell from a Gaussian Process (GP). GP were chosen as there has been research showing that they are good tools for learning spatiotemporal models such as wind from Sheahan et al. [13] and RF [2]. Gaussian Processes are a nonparametric data-driven Bayesian method for non-linear regression [12]. The first step of a Gaussian Process is to learn the hyper-parameters for a continuous probability density function from a finite set of observations. Using the learned model parameters, the GP predicts the joint distribution for the entire space by determining the value at new points with conditional probability.

For in-situ architecture, the observations are taken as the dimensional positions along with the Path Loss (backed out of the Received Signal Strength). In this research, both the total Path Loss and Correction Model are desired and thus two different GPs are setup. $PL = f_{pl}(p)$ and $EMF = f_{EMF}(p)$. One of the most important part of the GP framework is the kernel function (covariance function). The most common kernel function is the Squared Exponential shown in Eq 1

$$k(X_1, X_2) = \sigma_f^2 \; exp(\frac{-(X_1 - X_2)^2}{2l^2}) \qquad (1)$$

where the hyper parameters are $\theta = [l \; \sigma_f^2]$. The first hyper-parameter $l$ is the characteristic length scale and $\sigma_f^2$ is known as the signal variance. Learning the hyper-parameters is time consuming and is related to the number of data samples collected. In order to account for the smaller processing units onboard, the learning and prediction steps would be done at a much lower rate like once a minute. The prediction from the GP can be stored as a table from which communication aware applications can use along with the prediction from SPLAT!. There could be additional ways to increase the speed at which learning is performed such as sparsifying the measurements into the grid structure but the effects of these on performance is not known.

## III. SIMULATIONS

For simulations, the architecture was extended to act as the 'truth' data. SPLAT! output was used as the baseline and then a randomly generated correction field was applied

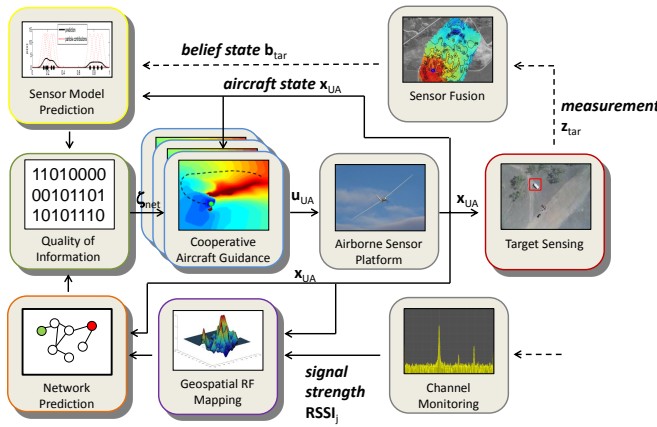

Fig. 1. Framework for communication-aware robot planning.

to create a simulated true mean field. A noise layer was added on top of this based on LOS characteristics and Doppler spectrum to simulate multipath effects and other sensor errors. Two different random models were used to for the generated correction field models. Experiments were conducted evaluating the capability of the Gaussian Process to learn the true mean field and the randomly generated correction field. For simulations, MATLAB's environment was used along with their machine learning and statistical toolbox for the Gaussian Process through *fitrgp* and *predict*.

*Correction Field Generation*

The first model adds an independent Gaussian random variable to each cell. The parameters for the Gaussian random variable come from Kasampalis et al. [5] and are set as $\mu$ = 13.4 and $\sigma$ = 7.1. These parameters are acceptable are the worst reported values from SPLAT! in the paper even if it was for another wavelength. Since there is no correlation between cells and transmitter location this would be a worst case scenario for the error field. The second model takes the first model then adds a level of correlation between the cells for the variation and mean. Intuitively this makes sense as RF propagation is correlated to terrain and areas near each other should experience similar levels of modelling errors. Equation 2 is the scaling factor for the variance and mean based on the distance from the transmitter. $d_{max}$ is the max distance away possible from the transmitter and by scaling this value by $b$ a reasonable result can be generated.

$$SF = \frac{d}{d_{max} * b} \qquad (2)$$

Figures 2 show the results from these two models with a transmitter centered in the middle of the grid structure.

*Measurement Model*

Lastly, measurement noise is added on top of the mean field generated by SPLAT! and the correction field representing multipath effects and sensor noise. Multipath fading is correlated to the antenna parameters (omni vs directional), relative motion of the transmitter/receiver and the line of sight characteristics. Two common statistical distributions have been used to model multipath fading; Rician if there is a LOS component and Rayleigh with no LOS. The parameters of these two distributions are related to the Doppler spectrum. The method used to generate these distributions is the sum of sinusoids. The approach taken is outlined in Rappaport [11]. We generate two frequency components for the wave based on the carrier frequency $w_c$ and the maximum Doppler frequency $w_d$ and perform a phase shift based on the angle of arrival from a Uniform distribution where $F_c$ is the carrier frequency and $V_{rel}$ is the relative velocity.

$$w_d = \frac{2\pi V_{rel} F_c}{c} cos('U'(0, 2\pi)) \qquad (3)$$

$$w_c = 2\pi F_c \qquad (4)$$

$$ray = \sum_{i=1}^{N} acos((w_c + w_d)t +' U'(0, 2\pi)) \qquad (5)$$

where $a$ is generated from a Weibull distribution with a $\sigma$ of 3. The difference between the Rayleigh to the Rician is that one component dominates and has no phase shift. The number of paths was chosen to be 16 for these simulations.

$$rice = 4.5cos((w_c+w_d)t)+\sum_{i=1}^{N-1} acos((w_c+w_d)t+'U'(0, 2\pi)) \qquad (6)$$

Using these two sum of sinusoids, they are demodulated using Quadrature amplitude modulation to get the imaginary and real components of the signal. By taking the norm of these and converting to dB the fade is generated for a set number of time samples on the time scale of the frequencies. When a measurement is taken, one sample is grabbed from the

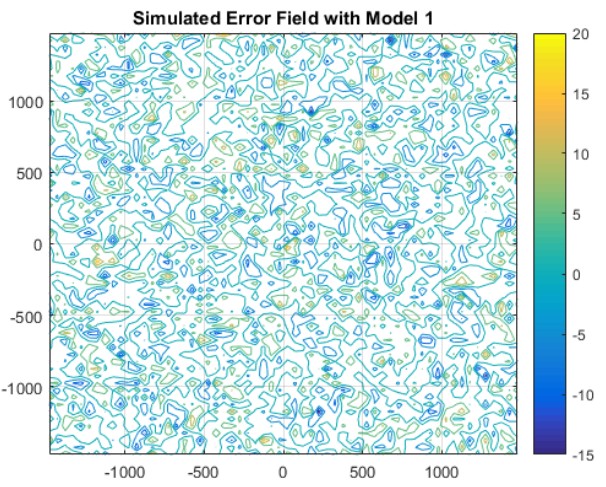
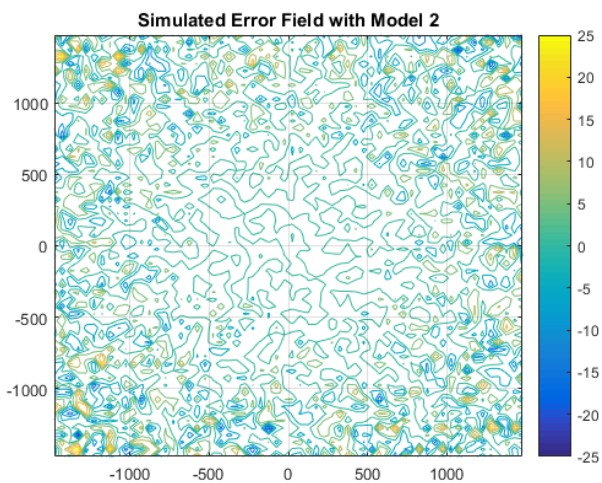

Fig. 2. Simulated error correction fields using model 1 (left) and model 2 (right)

| Parameter | MRS |
|---|---|
| Earth Dielectric Constant (relative permittivity) | 13.00 |
| Earth Conductivity (average ground) | .002 |
| Atmospheric Bending Constant | 301.00 |
| Frequency in MHZ | 2480 |
| Radio Climate | 5 |
| Polarization | 0 |
| Fraction of situations | .9 |
| Fraction of time | .9 |

TABLE I
LIST OF THE SPLAT! PROPAGATION CONFIGURATION PARAMETERS

generated time samples. Additional noise is added on top of the multipath fade as sensor errors randomly generated with a mean of .5 dB.

*Location*

Only a single region of interest will be presented here but the results are similar across multiple regions that were tested. The region that was chosen is the CU Mountain Research Station (MRS) located at 40.0331548, -105.5367712. This area is a mix of heavily wooded areas with large elevation changes. This provides a reasonable environment that represents conditions that emergency fire fighting crews might experience in the rural parts of Colorado Rockies. For the simulations, the transmitter is centered at the reference GPS location with an antenna height of 2 meters and a 3km square is used to constrain the region of interest. A simple aircraft model is used to fly a lawn-mower pattern with 200 meter E-W spacing with an extra 100 meter buffer. The aircraft model is flying at constant AGL which the aircraft will 'immediately' perform altitude changes if a change in elevation occurs. This is a reasonable assumption for now due to possible sensor requirements and flight restrictions. The RSSI measurement sample rate is set at 2 HZ and the GP components will be ran after the entire flight path is flown. The SPLAT! configuration for this region is listed in Table I

The Fraction of Situations and Time are both tunable pa-

rameters to determine the statistics of the median value output from the software and what was received. Time has to do with atmospheric fluctuations while Situation is position based. The higher the percentage the less variability in the model. The choices for Earth Dielectric Constant and Conductivity is based on table provided in SPLAT!'s documentation. These are four main tunable parameters to tweak the output to better fit the terrain.

## IV. RESULTS

The results for the Mountain Research Station simulation are shown here and are similar across the other regions that were analyzed. The main focus of this paper is investigating the difference in SPLAT! resolution, SPLAT! to Free Space Loss, and the capability of the Gaussian Processes to learn the true mean field and generated corrected field.

### A. DEM Resolution Comparison

The Figures in 3 and 4 show significantly different results. While not direct comparisons since the higher resolution has more grids, the shape of the top half of the region is not very similar at all. It is likely that the SRTM data with 3 arc second resolution has already perform some averaging across the terrain space and is resulting in incorrect terrain model. At this point, with out truth data to validate it is recommended to use the 1 Arc Second data when available.

### B. Simple Path Loss Comparison to SPLAT!

Figure 5 shows that the largest areas are a majority of the space that the two models were propagating in. The areas of the closest performance of Free Space Loss and SPLAT! Path Loss is when there is no longer LOS to the aircraft, the signal loss is closest to FSPL. Free Space Loss attenuates the signal significantly more than what SPLAT! predicts suggesting that FSPL is a worst case scenario for losses. This is not surprising as it would be expected that the ground plane would provide

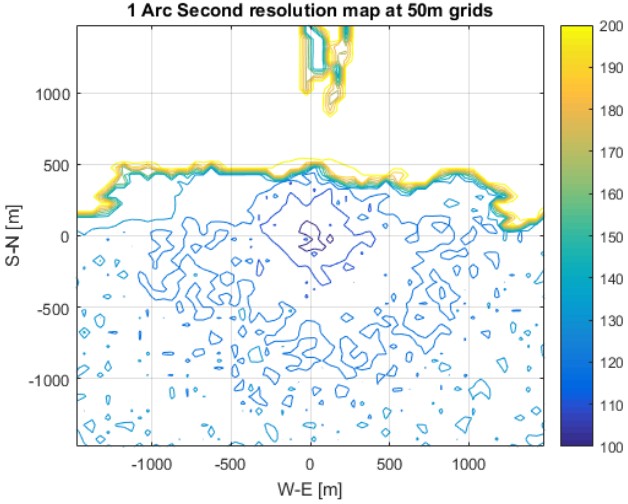

Fig. 3. SPLAT! output for 1 arc second resolution DEM for the MRS region

Fig. 5. The NOAA RoI (left) and MRS RoI (right) error between Free Space Loss and SPLAT! Path Loss

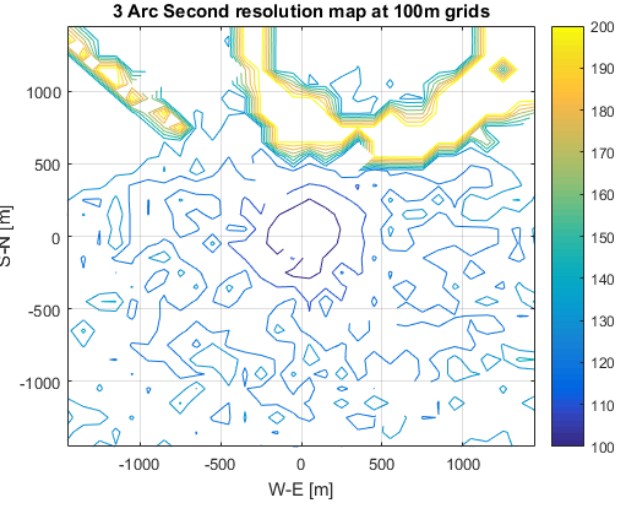

Fig. 4. SPLAT! output for 3 arc second resolution DEM for the MRS region

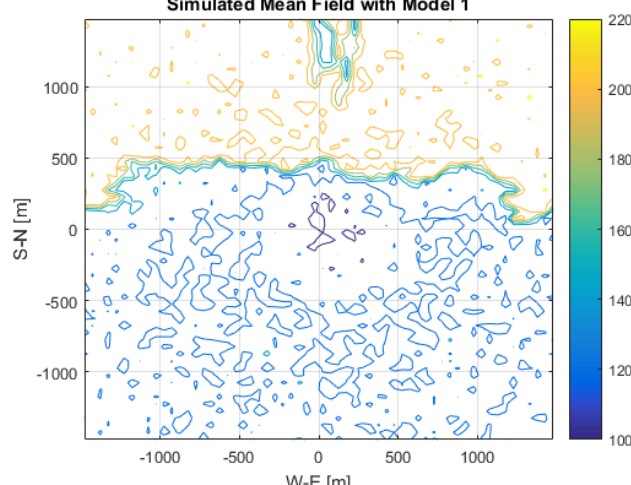

Fig. 6. Simulated mean path loss field from SPLAT and correction field with model 1

some constructive wave interference at distances away related to the wavelength of the signal.

### C. Simulated Mean Fields

Figures 6 and 7 illustrates the two simulated mean fields from SPLAT! prediction and the correction model using method 1 and 2. The key difference between the two models is that model 2 has more variation away from the center while the first model is consistently random all across. The correction models with the current parameters do not affect the overall shape of the path loss field.

### D. GP Predicted Error Fields

Figures 8 and 9 are the GP predicted correction fields with figures 10 and 11 being the difference of the GP predictions and the simulated truth values for the correction fields. The

prediction for the first model resembles a randomly generated process of noise however the second one only shows the gradual trend of the correction field losing some of the noise. This is an interesting results because it predicts how the correlation changes with position outwards but on other runs it looks more like the results for model 1. This is likely a function of the mulitpath effects which is different for every run. Unfortunately, neither of the models do a very good job of predicting the values for the field. The mean error for model 1 is 35.4 dB with a variance of 73.6 dB. For model 2 the mean error is 37.8 dB with a corresponding 188.3 dB variance. The mean is significantly higher than expected and is likely a result of the way the error measurements are taken without filtering any of the multipath effects. In addition it is very likely that

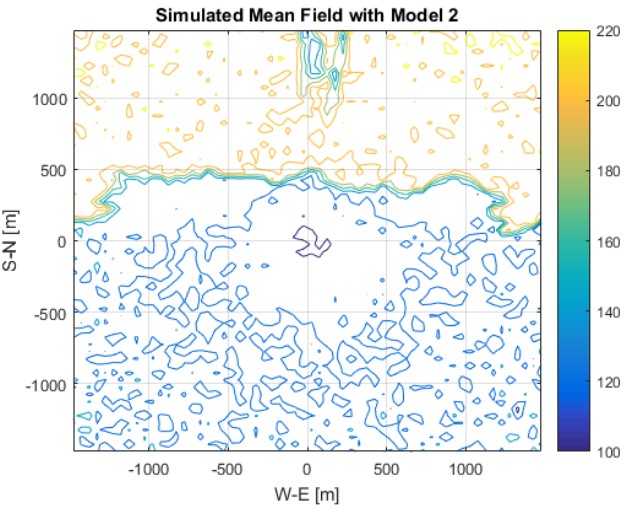

Fig. 7. Simulated mean path loss field from SPLAT and correction field with model 2

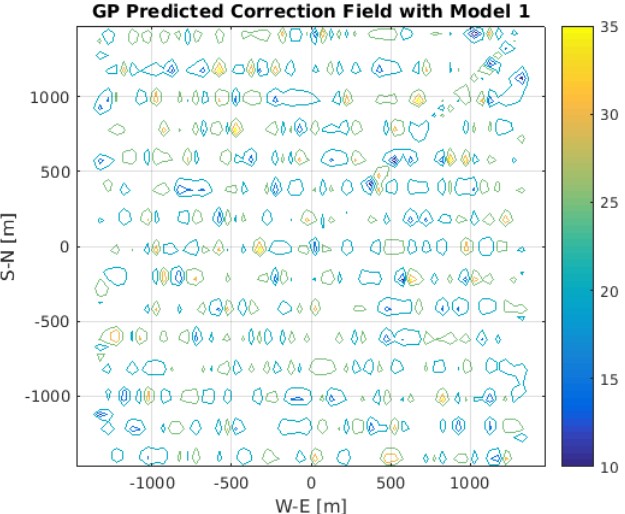

Fig. 8. GP prediction of the error correction field from simulated field with model 1

the large errors correspond to how sparse the data is sampled. Only a small number of locations are actually measured and thus the unmeasured errors are highly erroneous and skew the data. Finding this correction field may be more difficult than anticipated without high sample sets.

### E. GP Predicted Mean Fields

Figures 12 and 13 are the true mean field predictions using the GP while figures 14 and 15 are the error plots between the GP prediction and truth. Comparison of the structure for the mean field resembles that of the GP prediction. The main parts that had errors are the boundaries where the signal drops off. The error grows more so in between the path were measurements were collected. The mean error for model 1 is

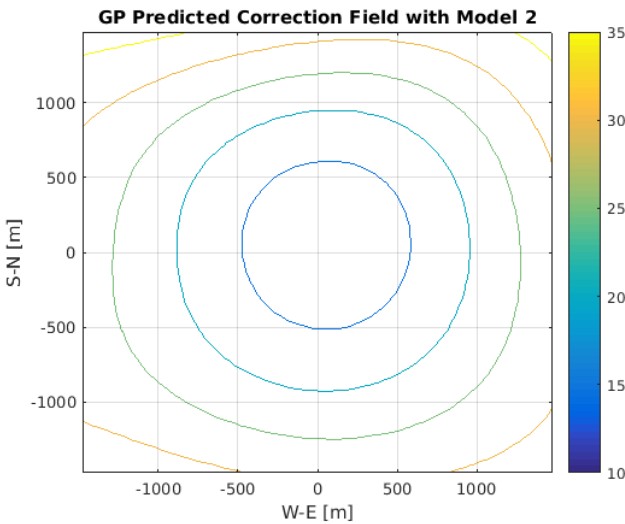

Fig. 9. GP prediction of the error correction field from simulated field with model 1

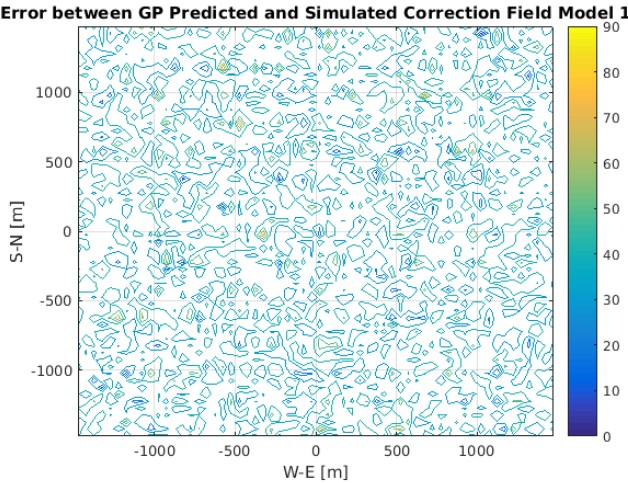

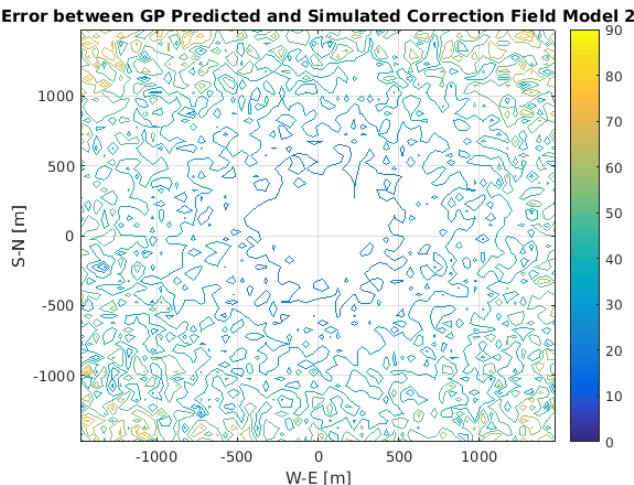

Fig. 11. Error between the GP prediction and simulated values for the error correction field with model 2

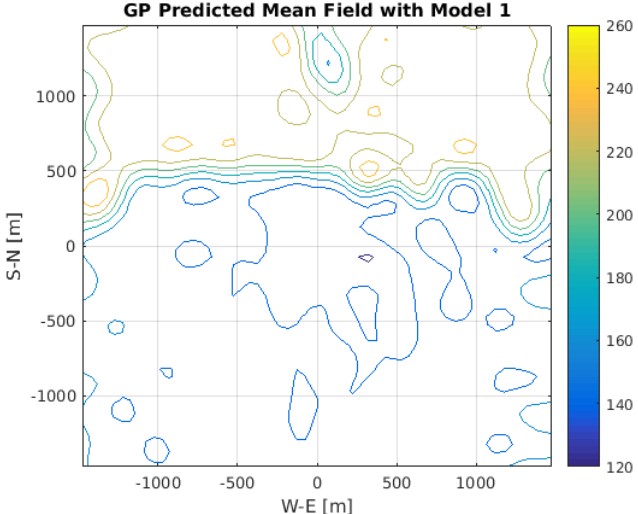

Fig. 12. GP prediction of the mean field from simulation with model 1

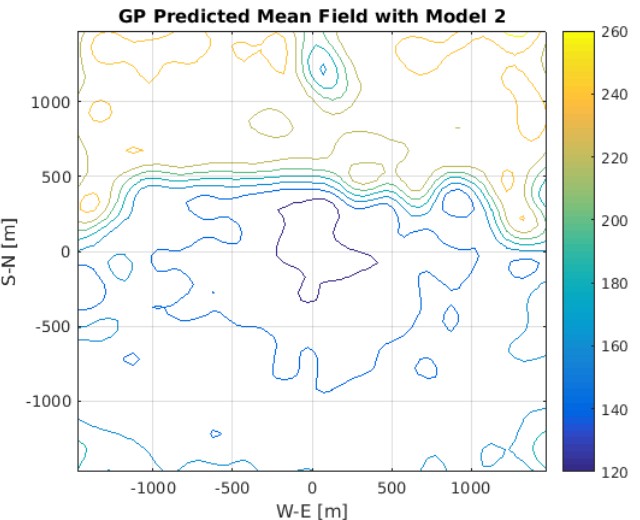

Fig. 13. GP prediction of the mean field from simulation with model 2

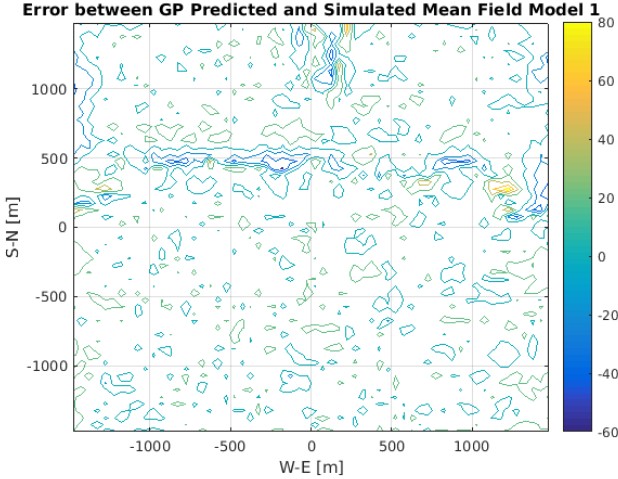

Fig. 14. Error between the GP and Simulated mean fields for model 1

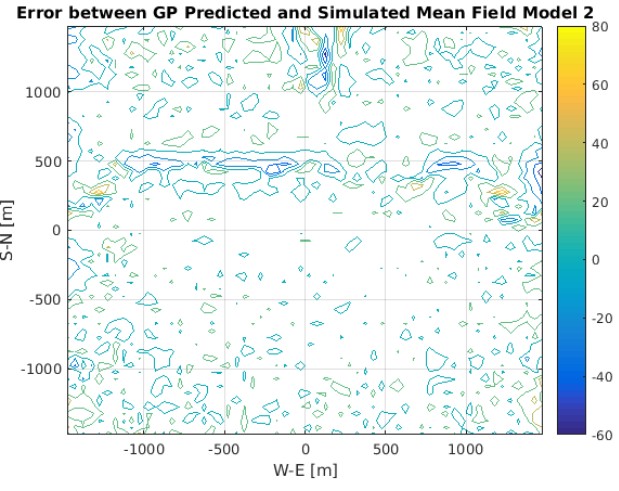

Fig. 15. Error between the GP and Simulated mean fields for model 2

7.54 dB with a variance of 166 dB while model 2 had a mean of 7.7 dB error and 169 dB variance. The correction field does not have a major effect on the overall performance of the GP to find the structure of the RF Field. It would be expected that with filtering to remove any of the noise from the simulated multipath effects the mean error could be decreased further.

## V. CONCLUSION

This paper describes an architecture that can be used for simulation and in-situ learning of the attenuation of RF signals in the environment. The architecture consists of two components, a prediction mean field and a correction mean field. The prediction mean field is generated using SPLAT!'s modified ITM. The correction mean field is learned online with a Gaussian Process. For simulation purposes the correction field is generated with simple models when no data is present. In addition measurement sensor noise is added to the simulation model accounting for multipath affects.

A single region of interest at CU Mountain Research Station is presented. Comparisons of the two different DEM resolutions for SPLAT! from SRTM is performed. While no truth data is available to verify these results, it is recommended to use the high res maps whenever possible. A comparison of SPLAT!'s output with Free Space Loss demonstrates that SPLAT!'s path loss values are lower than just free space loss as a result of the terrain affects. Lastly, the simulated mean field and results from the GP predictions are compared for both the correction field and mean path loss field. The results suggest that accurately finding the correction field will be difficult without large sample sets however it does provide a general overview of the structure. The error for the GP predictions of

the correction field were around 30 dB mean error. The GP did a significantly better job at learning the true mean field with the means around 7 dB. The GP learned structure looked very similar to the truth data with the largest areas in places that no samples were taken and the truth data had large variations. It is believed that by adding in a filtering technique to remove the time dependent noise, the means for both GP learning methods would be decreased. Further investigation is needed in seeing how the poor mapping of the correction field performs for use in Communication-aware applications or if additional modifications to the architecture is needed. Preliminary results in this area (not shown in the paper) indicate that the correction field error does not hinder the architecture's performance for communication applications.

## VI. Future Work

The main two focus areas of continuing this work is collecting data of actual RF path loss fields in several regions that CU Boulder has COAs available for. This data collection will occur summer and fall 2017. Using this data, the simulation correction models will be refined to more accurately resemble true environments. These flight tests will be conducted with fixed wing and multirotor vehicles in order to characterize the noise effects between stationary and mobile nodes. The second area of focus is to continue developing the simulation model features by adding in signal-to-noise ratios, frequency selective fading, bit rates, and other channel components. Part of this includes more advanced vehicle dynamics and adding a 3rd dimension to the models to fly ASL rather than AGL. This simulation model will than be applied to specific Communication-aware applications of interest to the researchers at RECUV including RF tracking, MANET, and communication relaying.

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
