# OpenReview forum: "Hybrid RF Propagation Model using ITM and Gaussian Processes for Communication-Aware Planning"
_roboticsfoundation.org/RSS/2017/RCW_Workshop/-_Proceedings_

### Review · AnonReviewer2 · 2017-06-26
**Paper review: accept**

**Rating:** 4
**Confidence:** 2

**Review:**


This paper presents a new method for modeling communication in multi-robot scenarios. The topic fits the topic of the workshop well, bringing the perspective of communication in real-world settings.

The general communication architecture is realistic (allowing for modeling of loss, noise, etc.), while still being general.
The paper would be strengthened by including more motivation for the model (why is this the right architecture?) and parameter choices (how can the parameters can be tuned properly?).

The experimental results are conducted in realistic mountainous domain, but some of the conclusions are unclear. More detail and additional analysis would be helpful.

The writing is generally clear, but there are several typos and some unclear text. For instance, it isn't always true that "agents must maintain links between each other while solving the task in order to gain a benefit over a single system" as this is highly dependent on the domain and communication needs. The paper should be polished before inclusion in the workshop proceedings.

---

### Review · AnonReviewer1 · 2017-06-28
**The presented idea is certainly valid, relevant to this workshop, and promises future research directions. In it’s current form, the paper would benefit from a clearer discussion of the results and model choices.**

**Rating:** 3
**Confidence:** 2

**Review:**

This work addresses the topic of communications modeling as a means for better robotic planning. In particular, the authors propose a framework that combines classical propagation models (providing a base line) with an adaptive model (providing the opportunity to learn from data, and is given by a Gaussian Process).

The presented idea is certainly valid, relevant to this workshop, and promises future research directions. In it’s current form, the paper would benefit from a clearer discussion of the results and model choices.

Can the authors comment on the choice of the kernel function of the Gaussian Process? It seems to me that this will dictate the correlation between collected radio measurements. Also, the authors should relate to literature in the robotics domain (Fink et al.) that uses GPs for learning RF models.
In this context, it would be interesting to discuss the practicality of the GP approach for very large online data sets (aircraft flying fast and sampling dense data).

Minor comments.
- Elaborate the acronym sUAS